# Biochemical Characterizations of the Putative Endolysin Ecd09610 Catalytic Domain from *Clostridioides difficile*

**DOI:** 10.3390/antibiotics11081131

**Published:** 2022-08-20

**Authors:** Hiroshi Sekiya, Hina Yamaji, Ayumi Yoshida, Risa Matsunami, Shigehiro Kamitori, Eiji Tamai

**Affiliations:** 1Department of Infectious Disease, College of Pharmaceutical Science, Matsuyama University, 4-2 Bunkyo-cho, Matsuyama 790-8578, Ehime, Japan; 2Research Facility Center for Science and Technology, Faculty of Medicine, Kagawa University, 1750-1 Ikenobe, Miki-cho, Kita-gun, Takamatsu 761-0793, Kagawa, Japan

**Keywords:** *Clostridioides difficile*, endolysin, antimicrobial agent, antimicrobial resistance

## Abstract

*Clostridioides difficile* is the major pathogen of pseudomembranous colitis, and novel antimicrobial agents are sought after for its treatment. Phage-derived endolysins with species-specific lytic activity have potential as novel antimicrobial agents. We surveyed the genome of *C. difficile* strain 630 and identified an endolysin gene, Ecd09610, which has an uncharacterized domain at the N-terminus and two catalytic domains that are homologous to glucosaminidase and endopeptidase at the C-terminus. Genes containing the two catalytic domains, the glucosaminidase domain and the endopeptidase domain, were cloned and expressed in *Escherichia coli* as N-terminal histidine-tagged proteins. The purified domain variants showed lytic activity almost specifically for *C. difficile*, which has a unique peptide bridge in its peptidoglycan. This species specificity is thought to depend on substrate cleavage activity rather than binding. The domain variants were thermostable, and, notably, the glucosaminidase domain remained active up to 100 °C. In addition, we determined the optimal pH and salt concentrations of these domain variants. Their properties are suitable for formulating a bacteriolytic enzyme as an antimicrobial agent. This lytic enzyme can serve as a scaffold for the construction of high lytic activity mutants with enhanced properties.

## 1. Introduction

*Clostridioides difficile* is a gram-positive, spore-forming, and anaerobic bacterium that causes infections leading to antibiotic-associated diarrhea, pseudomembranous colitis, and *C. difficile*-associated diarrhea [1]. For severe *C. difficile* infections, vancomycin or metronidazole are used as first-line treatments [2], but these adversely affect the gut microbiota. Fidaxomicin, a narrow-spectrum antibiotic [3], is another option that is less toxic to obligate anaerobic commensal bacteria, but its high cost limits its clinical use [4]. In addition, fecal microbiota transplantation [5], probiotic therapy [6], and monoclonal antibody treatment are available as alternative options to antimicrobial agents. However, each of these methods still has issues to be resolved and reasons why they cannot be easily introduced [7], so specific antimicrobials targeting *C. difficile* are required.

Lytic enzymes can kill bacteria by hydrolyzing the peptidoglycan of bacterial cell walls via their catalytic domains. Despite performing the same peptidoglycan cleavage function, lytic enzymes have significantly different structures and underlying mechanisms of action [8]. They are divided into four classes: glucosaminidases, muramidases, amidases, and endopeptidases, depending on the hydrolyzing sites. Autolysin and endolysin are well-known lytic enzymes. Bacterial endogenous autolysins are involved in different physiological functions that require bacterial cell wall remodeling, such as cell wall expansion, peptidoglycan turnover, daughter cell separation, sporulation, germination, peptidoglycan recycling, and/or autolysis [9,10,11]. Phage-derived endolysins are expressed in the final stage of infection to hydrolyze cell wall peptidoglycans, which facilitates bacterial lysis and progeny phage release [12]. In general, phage endolysins exhibit species-specific lytic activity [13]. The molecular mechanisms of this species specificity are not well understood, but some are derived from the binding domain [14,15] and others from the structure of the substrate binding site of the catalytic domain [16]. Especially in *C. difficile*, the peptide bridge of its peptidoglycan is composed of amino acids that are different from those of other bacteria, and its structure is also thought to be different. The peptide bridge of *C. difficile* has been reported to be composed of the following four amino acids: L-Ala-γ-D-Glu-mDAP-D-Ala (mDAP: meso-2,6-diaminopimelic acid), and the glycan backbones are connected by direct 3–4 cross-links [17]. Therefore, autolysins generally have no species specificity, but the autolysin in *C. difficile*, Acd24020, which has endopeptidase activity, has species specificity, and we have proposed a particular mechanism for this [16].

Lytic enzymes, especially those with species-specific lytic activity, are potential therapeutic agents that can provide an alternative to standard drug therapy [18,19,20]. In group A *Streptococcus* infections, an endolysin specific for this organism has been shown to be effective in vivo when applied directly to the site of infection [21]. However, for a protein to be used as a drug, it should have the properties of being able to be purified in large quantities in a solubilized state, be resistant to protease degradation, and be suitable for formulation. For example, in intestinal infections of *C. difficile*, a therapeutic enzyme must be delivered to the intestinal tract without the loss of enzymatic activity. Enteric capsules and coatings make this possible, but to adapt these technologies, it is important than an enzyme does not lose its activity upon lyophilization.

In a survey of genes of the *C. difficile* strain 630, we found an endolysin gene (gene ID: CD09610). The corresponding protein, named Ecd09610 (endolysin of *C. difficile* gene ID CD09610), has glucosaminidase and endopeptidase catalytic domains at the C-terminal end belonging to the GH73 family and NlpC/P60 family, respectively. In addition, the gene CD29030, which has the same sequence as CD09610, is present in *C. difficile* 630, and there are many genes related to phage proteins around these genes. Mondal et al. discovered a cell wall hydrolase (CWH) lysin encoded in the genome of the *C. difficile* phage phiMMP01, which is similar to this endolysin, and characterized its properties, including the species specificity of its two successive catalytic domains [22]. However, there are few reports of glucosaminidase endolysins of the GH73 family [23,24], most of which are autolysins [25,26,27,28], and only some of which have even been structurally characterized [29,30,31,32,33,34]. In contrast, the endopeptidases of the NlpC/P60 family are found in endolysins and autolysins [35,36,37], some of which have been structurally characterized [38,39,40].

Here, we report the detailed biochemical properties of each of the two catalytic domains of Ecd09610, including their optimal pH and salt concentrations and the effects of metal ions, temperature, and lyophilization.

## 2. Results

### 2.1. Identification, Cloning, Expression, and Purification of Ecd09610 and Its Derivatives

Gene ID CD09610 in the *C. difficile* 630 genome was identified as a putative endolysin by sequence similarity searches of glucosaminidase and endopeptidase, and the corresponding protein was named Ecd09610 (endolysin of *C. difficile* gene ID CD09610). This gene has an uncharacterized domain in the N-terminal region and a glucosaminidase domain (Pfam number: PF01832) and endopeptidase domain (Pfam number: PF00877) in the C-terminal region (Figure 1a). Residues essential for catalysis (Cys, His, Asp) in the NlpC/P60 family of endopeptidases are also conserved in Ecd09610 (Appendix A). The genes encoding Ecd09610 and its domains were cloned into an *Escherichia coli* expression vector (pColdII) that is able to fuse a His-tag at the N-terminus. The entire region (Ecd09610), the glucosaminidase domain (Ecd09610CD53), the endopeptidase domain (Ecd09610CD3), and the C-terminal domain with both catalytic domains (Ecd09610CD1) were successfully expressed and purified by Ni-affinity chromatography (Figure 1b). In the purification of Ecd09610CD53, Ecd09610CD3, and Ecd09610CD1, approximately 11.6 mg, 3.0 mg, and 3.7 mg, respectively, were recovered from 400 mL of culture. However, in the purification of the wild-type, the purification itself was performed as successfully as the purification of each domain variant, but 40% of the purified protein was precipitated by dialysis and only 60% was recovered (1.1 mg/400 mL culture). In contrast, for Ecd09610CD53, Ecd09610CD3, and Ecd09610CD1, the protein loss on dialysis was 10%. Taken together, the solubility of Ecd09610 was significantly improved by dividing it into domain variants. To examine the lytic and binding activity of the purified proteins, turbidity reduction assays and binding assays were performed using *C. difficile* 630 as a substrate. As shown in Figure 1c, Ecd09610 and each domain variant had lytic activity. Compared with the wild-type Ecd09610, Ecd09610CD3, which has only the endopeptidase domain, showed almost the same activity, while Ecd09610CD53, which has only the glucosaminidase domain, showed lower lytic activity, and Ecd09610CD1, which has both catalytic domains, showed the highest activity. Furthermore, binding activity assays of the wild-type and each domain variant against *C. difficile* 630 showed that the catalytic domain itself can bind to the cells (Figure 1d).

### 2.2. Characterization of the Lytic Activity of Ecd09610 and Its Derivatives

To characterize the enzymatic activity of Ecd09610 and its domain variants, we determined the effects of pH, salt, metal ions, and temperature on the lytic activity using *C. difficile* 630 as a substrate (Figure 2a–d). As shown in Figure 2a, the optimal pH of wild-type Ecd09610, Ecd09610CD1, and Ecd09610CD53 was 6, while Ecd09610CD3 had its highest lytic activity at a pH of 8. Note that at pH 8, Ecd09610CD3, which is only an endopeptidase, showed high lytic activity, whereas the activity by the endopeptidase was counteracted in Ecd09610CD1 and Ecd09610, which have two catalytic domains. This phenomenon is seen at pH 7 and above. At pH 6 and 5, glucosaminidase and endopeptidase work together, and the two domains of Ecd09610CD1 show additive lytic activity.

The effects of salt (NaCl) concentrations on the lytic activity of Ecd09610 and each domain variant were also examined (Figure 2b). Ecd09610 and Ecd09610CD53 showed high lytic activity below 50 mM, decreased activity above 50 mM, and almost no activity above 150 mM. In Ecd09610CD1 and Ecd09610CD3, the lytic activity was highest at 75 mM. Moreover, the activity of Ecd09610CD1 was reduced above 75 mM, while that of Ecd09610CD3 was not reduced up to 200 mM.

In addition, the effects of metal ions were examined (Figure 2c). The addition of CaCl_2_ and ethylenediaminetetraacetic acid did not significantly affect the lytic activity, but the addition of MgCl_2_ slightly increased activity, and the addition of ZnCl_2_, MnCl_2_, and CuCl_2_ significantly reduced the lytic activity.

To clarify the thermal stability of each domain variant, their activities were measured with purified proteins that were preincubated for 10 min at varying temperatures (Figure 2d). The results showed that the activities of Ecd09610CD1 and Ecd09610CD3 decreased significantly as the temperature increased, but maintained some activity above 60 °C. However, Ecd09610CD53, which contains only the glucosaminidase domain, had low activity, but showed no decrease in activity up to 100 °C.

### 2.3. Bacterial Specificity of Ecd09610 Derivatives

In general, phage endolysins exhibit species-specific lytic activity [13]. Therefore, the species specificity of each domain was determined by turbidity reduction assay using various bacterial strains as substrates (Table 1 and Appendix A). Each of the three domain variants showed lytic activity against three strains of *C. difficile*. Ecd09610CD3 also showed weak lytic activity against *Clostridium histolyticum*, *Clostridium ramosum*, *Clostridium tetani*, *Atopobium fossor*, and *Bacillus subtilis*, but showed no lytic activity against the other gram-positive bacteria tested in this study. Ecd09610CD1 and Ecd09610CD53 showed weak lytic activity against *Clostridium novyi* in addition to the above-mentioned bacteria. The species specificity of binding was also tested (Appendix A). The three domain variants bound to most of the bacteria tested in this study, but Ecd09610CD3 and Ecd09610CD53 exhibited reduced binding activity against *Clostridium perfringens* and *Eubacterium cylindroides*.

### 2.4. Long-Term Stability of Ecd09610 Derivatives

For enzymes to be used as drugs, their stability, both long-term and under dry conditions, needs to be verified. Each purified Ecd09610 derivative was lyophilized, stored at either room temperature or 4 °C, and then dissolved again in water to determine its lytic activity (Figure 3). There was no loss of the activity due to lyophilization among the three domain variants. There was a slight decrease in the activity for Ecd09610CD1 when stored at 4 °C for 4 weeks, but no decrease in activity was observed for the other two domain variants. Furthermore, when stored at room temperature for 4 weeks, there was a decrease in activity of about 49% and 40% for Ecd09610CD1 and Ecd09610CD3, respectively, but no decrease in activity for Ecd09610CD53

## 3. Discussion

The gene ecd09610 (gene ID CD09610) was identified by a homology search for glucosaminidase and endopeptidase from the genome of *C. difficile* 630. This gene product is quite similar to the recently reported endolysin CWH of *C. difficile* phage phiMMP01 [22]. The similarity and identity between Ecd09610 and CWH in the N-terminal uncharacterized domain, the glucosaminidase domain, and the endopeptidase domain were 98% and 90%, 87% and 65%, and 99% and 99%, respectively (Appendix A). There were also genes associated with phages around the Ecd09610 gene (Appendix A). This suggests that a phiMMP01-like phage had infected and lysogenized in *C. difficile* 630.

The lytic activities of wild-type Ecd09610 and its domain variants against *C. difficile* 630 were measured, revealing that Ecd09610CD1 had the highest activity per molecule, wild-type and Ecd09610CD3 were comparable, and Ecd09610CD53 had the lowest (Figure 1c). These results are similar to those of Mondal et al. [22]. The catalytic domain may have high affinity for the substrate since each domain variant binds to *C. difficile* even without the N-terminal uncharacterized domain, which is the putative binding domain [22] (Figure 1d). In addition, Ecd09610CD1, a variant lacking the N-terminal uncharacterized domain, exhibited higher activity than the wild-type. We also consider that this domain is not a meaningful binding domain since Mondal et al. reported that this region has similarities with the bacteriophage tail protein [22]. To address whether domain variants without the putative binding domain are more active, we cloned and purified the N-terminal uncharacterized domain of Ecd09610 alone and performed cell binding assays against *C. difficile* 630. We found that the N-terminal uncharacterized domain binds slightly to *C. difficile* 630 (Appendix A). Taken together, these data suggest that the N-terminal uncharacterized domain may prevent the normal binding of the catalytic domain to *C. difficile* 630, thereby reducing its lytic activity. It will be necessary to clarify the function of this uncharacterized domain in the future.

The results for the optimum pH assays for each domain variant were complicated. Notably, the activity of the endopeptidase was highest at pH 8, but despite the presence of endopeptidase domains in Ecd09610CD1 and in the wild-type, their activity was greatly reduced at pH 8 (Figure 2b). This may be due to the binding of the glucosaminidase, which does not exhibit lytic activity at pH 8, to the cells, resulting in diminished binding of the endopeptidase to the cells and, consequently, reduced endopeptidase lytic activity. These data suggest that the substrate affinity of the glucosaminidase is higher than that of the endopeptidase. The most effective pH is pH 8 when the endopeptidase alone is used to lyse bacteria, and pH 6 when used with the glucosaminidase alone or with the endopeptidase. In addition, regarding the effect of salt, the decrease in activity of the wild-type, Ecd09610CD1, and Ecd09610CD53 at high salt concentrations (>100 mM) is thought to be due to the decreased activity and binding of glucosaminidase to the cells (Appendix A). However, *C. difficile*-infected patients suffer from severe diarrhea, and the colon, which is the site of *C. difficile* infection, is thought to be inflamed with fluid seeping into the intestinal tract, so the salt concentration at the site is thought to be as high as or lower than that of saline solution. Therefore, the endolysin, which acts at low salt concentrations, should be effective. The results of the temperature tolerance study show that, interestingly, the Ecd09610 domain variants are relatively resistant to high temperatures, especially Ecd09610CD53, which was not inactivated by heating at 100 °C for 10 min, and the activity of Ecd09610CD1 above 45 °C seems to come from the glucosaminidase domain. Whether this thermostability is due to the speed of refolding or to heat tolerance of the protein itself remains unclear, but we intend to clarify this issue through further structural analysis. Furthermore, the domain variants of Ecd09610 do not lose activity by lyophilization and can be stored for long periods at 4 °C in a dried state (Figure 3). These properties may be advantageous for formulation as an antimicrobial agent. Moreover, it is difficult to obtain the purified wild-type endolysin in large quantities; however, the purified domain variants can be easily obtained in large amounts in soluble form. This property of the domain variants is also an advantage in industrial production.

The domain variants of Ecd09610 had weak lytic activity against some bacteria, but it is basically a *C. difficile*-specific endolysin (Table 1). The mechanism of the species specificity is known to depend on the binding domain [14,15] or on the structure of the substrate recognition groove [16]. In the case of Ecd09610, its substrate specificity mechanism is thought to be the latter, i.e., dependent on the structure of the substrate recognition groove since various domain variants of Ecd09610 exhibited specific lytic activity against *C. difficile* even though they bound to a variety of bacterial species without the binding domain. To elucidate the species-specificity mechanism, we intend to perform structural analysis of each domain. This specificity for *C. difficile* could be used to develop antimicrobial agents that can treat *C. difficile* infectious diseases, such as pseudomembranous colitis, without affecting the intestinal microbiota. However, the issue for using Ecd09610CD53 as an antimicrobial agent is its low lytic activity, so it may be necessary to create mutants with increased lytic activity. To achieve this, it is necessary to determine the structure of the catalytic domain and clarify the amino acids constituting its active center. Then, mutants can be constructed by substituting the amino acids so that the active center structure will be more catalytically active. In addition, we would like to increase tolerance to proteases by amino acid substitutions based on structural information.

It has been reported that a synergistic effect can be achieved by combining lytic enzymes with different cleavage sites [41,42]. In Ecd09610CD1, each catalytic domain with a different cleavage site is linked, so we thought that they might act synergistically and that the linkage of the two domains might make sense. Therefore, we compared the lytic activity of Ecd09610CD1 with that of simultaneously added Ecd09610CD3 and Ecd09610CD53 (Appendix A). The results suggested that they seem to act synergistically in the initial phase of the reaction, but the final lytic activity was considered to be additive. No effect of the linkage of the two catalytic domains was observed. This result may be due to the low activity of the glucosaminidase and the differences in optimum pH and salt concentration of the respective catalytic domains.

## 4. Materials and Methods

### 4.1. Construction of Plasmids

The overlap extension PCR method [43] using Tks Gflex™ DNA Polymerase (TakaRa Bio, Inc., Shiga, Japan) was used for the construction of expression vectors for N-terminal His-tagged Ecd09610. PCR was performed using the primers listed in Appendix A and genomic DNA of *C. difficile* 630 as a template. The 2nd PCR products were digested with *Nde*I and *Bam*HI and then cloned into the expression vector pColdII (TakaRa Bio, Inc.) at the *Nde*I-*Bam*HI site. The resultant plasmid was designated as pColdIICD09610. The plasmids expressing both glucosaminidase and endopeptidase, glucosaminidase only, and endopeptidase only, (pColdIICD09610CD1, pColdIICD09610CD53, and pColdIICD09610CD3, respectively), were constructed by the same method using the primers listed in Appendix A and pColdIICD09610 as a template. PCR-amplified fragments in all constructs were verified with an ABI PRISM 3130xl genetic analyzer (Thermo Fisher Scientific, Waltham, MA, USA).

### 4.2. Preparation of Proteins

*E. coli* BL21-CodonPlus-RIL transformed with pColdIICD09610, pColdIICD09610CD1, pColdIICD09610CD3, or pColdIICD09610CD53 were cultured in M9 medium containing 0.2% (*w*/*v*) glucose, 0.2% (*w*/*v*) tryptone, 0.001% thiamine, 100 μg/mL ampicillin, 30 μg/mL chloramphenicol, and 10 μg/mL tetracycline at 37 °C until the middle-logarithmic phase, then incubated on ice for 30 min. After the addition of a final concentration of 1 mM isopropyl-β-D-thiogalactopyranoside, the cells were further incubated at 15 °C for 20–24 h. The harvested cells were suspended in buffer A (50 mM Tris-HCl, pH 7.0, 500 mM NaCl, and 20 mM imidazole), and the suspension was sonicated on ice for 30 s for a total of five times at power level 5 by an ultrasonic disruptor (UD-200, TOMY Co, Ltd., Tokyo, Japan). The suspension was then centrifuged at 22,300× *g* at 4 °C for 10 min, and the supernatant was filtrated with a 0.2-μm pore size syringe filter (Minisalt^®^, Sartorius, Göttingen, Germany). The protein solution was applied to an Ni^+^-charged Chelating Sepharose Fast Flow (GE Healthcare Bio-Sciences AB, Uppsala, Sweden). The column was washed with buffer A and then eluted by a stepwise gradient or a linear gradient of 50–350 mM imidazole. The elutant from the resin was dialyzed against buffer B (25 mM phosphate buffer, pH 6.0, 100 mM NaCl, and 10% glycerol) for CD09610 and buffer C (25 mM Tris-HCl, pH 7.0, 100 mM NaCl, and 10% glycerol) for the others, and filtrated with a 0.2-μm syringe filter.

### 4.3. Lytic Activity Assay

The lytic activity of proteins was tested by the method of Gerova et al. with some modifications [44]. Briefly, *C. difficile* strains cultured in TY medium and the other strains cultured in GAM medium at 37 °C for 16 h were washed twice and suspended in wash buffer (25 mM Tris-HCl, pH 7.0), then adjusted to 1.25 optical density at 600 nm (OD_600_)/mL. The lytic activity was started by the addition of 20 μL protein or assay buffer into 180 μL of a preincubated cell suspension. Their OD_600_ was measured at 37 °C for 1-min intervals (SpectraMax^®^ M5e Multi-Mode Microplate Readers, Molecular Devices Corp., Sunnyvale, CA, USA). In testing for thermal stability, samples were heated for 10 min and then left at room temperature for several minutes before measurement.

### 4.4. Cell Binding Assay

Cell binding assays were carried out with purified Ecd09610, Ecd09610CD1, Ecd09610CD3, and Ecd09610CD53. The purified protein and ovalbumin were incubated for 15 min on ice either with or without heat-inactivated cells in binding buffer (25 mM Tris pH 7.0) containing 100 mM NaCl (Figure 1d) or without NaCl (Appendix A). The samples were centrifuged at 22,300× *g* at 4 °C for 3 min. SDS-PAGE sample buffer was added to the supernatant, and the mixture was incubated at 95 °C for 5 min then analyzed by SDS-PAGE.

## Figures and Tables

**Figure 1 antibiotics-11-01131-f001:**
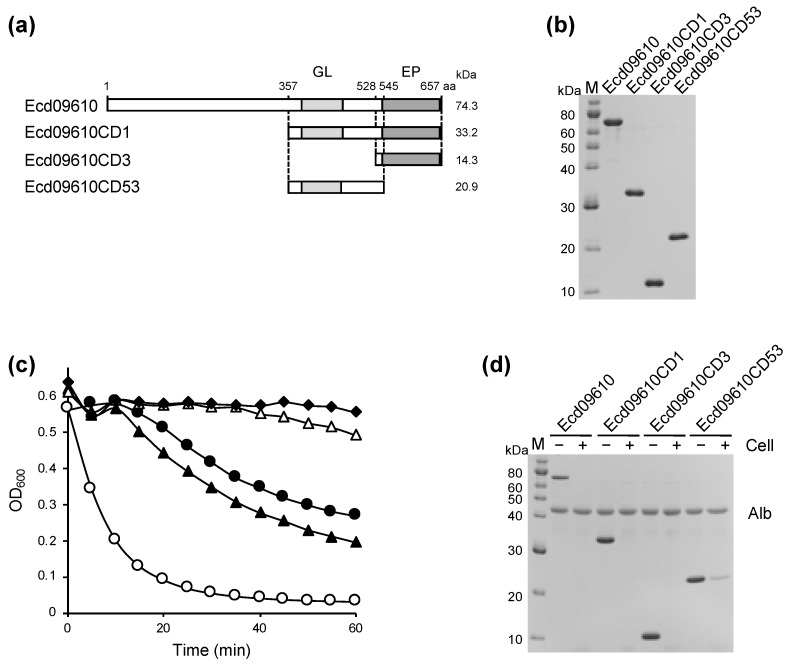
SDS-PAGE analysis, lytic activities, and binding ability of Ecd09610, Ecd09610CD1, Ecd09610CD3, and Ecd09610CD53. (**a**) Schematic diagrams of Ecd09610, Ecd09610CD1, Ecd09610CD3, and Ecd09610CD53. Ecd09610 has two catalytic domains of glucosaminidase (GL) and endopeptidase (EP) at the C-terminus. These proteins have His-tags at the N-terminus. (**b**) SDS-PAGE analysis of purified Ecd09610, Ecd09610CD1, Ecd09610CD3, and Ecd09610CD53 (1 μg each). The gel was stained with Coomassie blue R. (**c**) Lytic activities of protein (0.1 μM) were determined by the turbidity reduction assay against *C. difficile* 630 cells. Ecd09610 (filled circles), Ecd09610CD1 (open circles), Ecd09610CD3 (filled triangles), Ecd09610CD53 (open triangles), and control (filled diamond) are shown. (**d**) Binding ability of purified proteins to *C. difficile* 630 cells. Purified protein and nonbinding internal standard (ovalbumin: Alb) were incubated with (+) or without (−) cells. After centrifuging samples, supernatants were analyzed by 13.5% SDS-PAGE.

**Figure 2 antibiotics-11-01131-f002:**
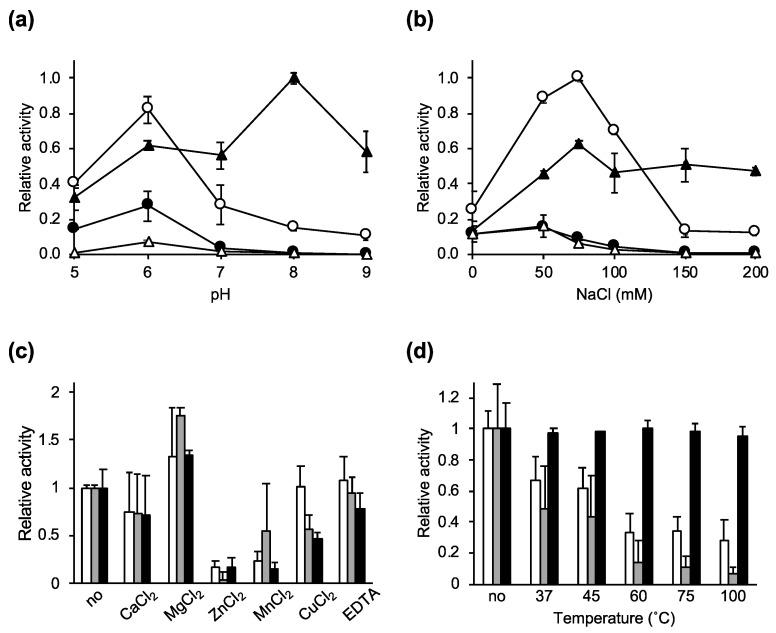
Lytic activity of Ecd09610, Ecd09610CD1, Ecd09610CD3, and Ecd09610CD53 against *C. difficile* 630 as determined by the turbidity reduction assay. The lytic activity was calculated after 30 min as follows: [ΔOD_600_ test (protein added) − ΔOD_600_ control (buffer only)]/µmol protein. (**a**) The optimal pH for lytic activity was determined using borate-phosphate universal buffer. The relative activity at pH 6.0 of Ecd09610CD3 was set as 1. (**b**) The effect of NaCl on lytic activity was determined using 25 mM Tris-HCl (pH 7.0). The relative activity at 75 mM NaCl of Ecd09610CD1 was set as 1. Ecd09610 (filled circles), Ecd09610CD1 (open circles), Ecd09610CD3 (filled triangles), and Ecd09610CD53 (open triangles) are shown. (**c**) The effects of divalent metal cations on lytic activity were determined by the addition of 1 mM CaCl_2_, MgCl_2_, ZnCl_2_, MnCl_2_, CuCl_2_, or EDTA. The relative activities are shown with activity in the absence of divalent cations set as 1. The lytic activity with no divalent cations was 35,011 (Ecd09610CD1, white), 16,788 (Ecd09610CD3, gray), and 11,190 (Ecd09610CD53, black). (**d**) The thermal stability of lytic activity was determined by measuring lytic activity after 10 min of heat treatment at 37, 45, 60, 75, or 100 °C, or with no treatment. The relative activities with no treatment for Ecd09610CD1, Ecd09610CD3, and Ecd09610CD53 were set as 1. The lytic activities with no treatment were 104,200 (Ecd09610CD1, white), 62,917 (Ecd09610CD3, gray), and 5288 (Ecd09610CD53, black). Means in all experiments were calculated based on three independent experiments. Standard deviations were calculated by three independent experiments, each with triplicate samples.

**Figure 3 antibiotics-11-01131-f003:**
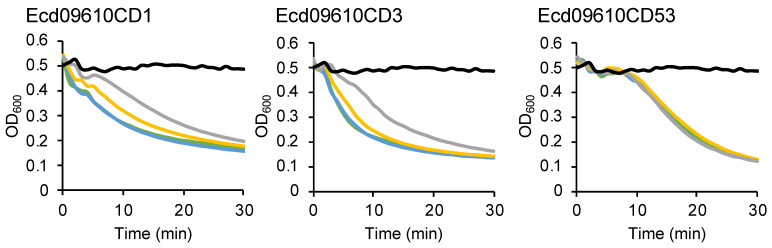
Effect of lyophilization on the lytic activity of proteins. Ecd09610CD1 (0.08 μM), Ecd09610CD3 (0.15 μM), and Ecd09610CD53 (0.3 μM) or buffer was added to the cells, and OD_600_ was measured at 1-min intervals for 30 min. No lyophilized protein (green), lyophilized protein (blue), lyophilized and stored at 4 °C, for 4 weeks (yellow), lyophilized and stored at room temperature for 4 weeks (gray), and buffer (black) are shown.

**Table 1 antibiotics-11-01131-t001:** Species specificity of Ecd09610 domain variant lytic activity.

Bacteria	Relative Activity (%)		
	Ecd09610CD1	Ecd09610CD3	Ecd09610CD53
*C. difficile* 630	100.0 ± 14.1	100.0 ± 20.8	100.0 ± 10.00
*C. difficile* ATCC43255	104.9 ± 9.00	97.6 ± 15.6	15.6 ± 5.20
*C. difficile* ATCC9689	101.7 ± 21.8	119.3 ± 21.0	96.3 ± 21.0
*C. acetobutylicum* ATCC824	−10.2 ± 0.90	−6.8 ± 4.10	−16.9 ± 4.90
*C. coccoides* ATCC29236	−1.3 ± 0.70	−0.8 ± 0.90	−2.3 ± 7.40
*C. histolyticum* JCM1403	12.6 ± 4.30	8.1 ± 2.10	5.8 ± 1.50
*C. lituseburense* ATCC25759	3.0 ± 5.10	4.5 ± 3.80	3.3 ± 1.50
*C. novyi* ATCC17861	22.4 ± 11.0	−2.3 ± 6.10	10.9 ± 17.40
*C. perfringens* strain13	5.6 ± 0.60	1.3 ± 3.50	3.7 ± 5.20
*C. ramosum* ATCC25582	65.3 ± 1.70	17.3 ± 2.90	7.5 ± 2.30
*C. tetani* KZ1113	34.8 ± 3.00	38.9 ± 1.00	11.0 ± 1.80
*A. fossor* ATCC43386	15.0 ± 2.20	11.0 ± 2.50	5.0 ± 2.00
*B. adolescentis* ATCC15703	0.3 ± 2.10	−2.4 ± 3.50	−4.9 ± 0.80
*E. cylindroides* ATCC27805	1.4 ± 5.00	−2.6 ± 2.20	−2.3 ± 2.60
*B. subtilis* ATCC6633	16.8 ± 3.20	8.9 ± 10.0	8.5 ± 3.90
*S. aureus* FDA209P	1.8 ± 1.90	2.6 ± 2.00	−1.1 ± 4.90

The lytic activity was calculated after 10 min as follows: {ΔOD_600_ test (0.14 μM Ecd09610CD1, 0.2 μM Ecd09610CD3, or 0.3 μM Ecd09610CD53 added) − ΔOD_600_ control (buffer only)}/μmol protein. The relative activity of the bacteria is shown with the lytic activity of *C. difficile* 630 set as 100%. Means and standard deviations in all experiments were calculated by three independent experiments each with triplicate samples. The lytic activities of *C. difficile* 630 [{ΔOD_600_ test − ΔOD_600_ control (buffer only)}/μmol protein] were 12,858 (Ecd09610CD1), 7957 (Ecd09610CD3), and 6102 (Ecd09610CD53).

## Data Availability

The data that support the findings of this study are available from the corresponding authors upon request.

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
