# Peer review of "Biochemical Characterizations of the Putative Endolysin Ecd09610 Catalytic Domain from Clostridioides difficile"

_antibiotics, 2022, doi:10.3390/antibiotics11081131_

Round 1

Reviewer 1 Report

This manuscript by Sekiya et al describes the identification and characterization of domains of an endolysin of Clostridioides difficile. 

Sekiya et al.

Biochemical Characterizations of the Putative Endolysin Ecd09610 Catalytic Domain from Clostridioides difficile

1. The abstract states that purified domains activity was very selective, targeting mainly C. difficile. The authors claim that the selectivity is due to selective cleavage of the substrate (Peptidoglycan, PG) than selective binding. In this regard, a scheme depicting the structure of PG of C.difficile and the differences with other clostridial species PG should be included. This turns out to be an important point for the general reader as differential endopeptidase activity can be related to aminoacid differences in the PG structure. Moreover, the authors cannot differentiate binding selectivity from catalytic activity differences without proper experimental design, i.e, a gfp fusion to address binding or enzymatic activity of each domain on purified PG.

2. The purification of the endolysin and its domains should be quantitated as endolysins are notoriously difficult to recover as soluble proteins, which could be a potential barrier to the proposed end of developing an antibacterial agent. Besides, several endolysins have been described as undergoing a proteolytic processing detection of which can be easily achieved by zymogram.

3. The effect of temperature on domain activity is not clearly described and needs to be improved. The authors state that purified domains are incubated at varying temperatures but there is no description of the timeline of the experiment, that is, how the sample is handled between the heating process and the biological activity determination. Domain refolding is probably taking place after the heating treatment. The authors should clearly differentiate thermal stability of the domain, that could be measured with a variety of physical techniques, from activity under thermal stress that should be measured as PG cleavage under varying temperature conditions

Minor points

Lines 274-276. “Degradation” should be replaced by “cleavage site”

Reviewer 2 Report

This is a valuable basic research. Clostridium difficile is a major causative agent of diseases including antibiotic-associated diarrhea, and patients are in the context of intestinal flora disturbances, which impose great limitations on the application of specific antibiotics and other antibacterial agents. Bacteriophages and their derived preparations are an effective means to effectively deal with specific bacterial infections in the future. 

Major revisions:

1. The authors have a good foundation in lyase and C. difficile research, such as citation 16. The authors can highlight their C. difficile research in the introduction.

2. The authors mainly analyzed the biochemical characteristics of the candidate domains, and discussed the effect of different environments (ion concentration, etc.) on the bacteriostatic ability. Since the infection site of C. difficile is relatively limited, can the authors specifically emphasize which concentration can correspond to the environment in which the bacteria are located?

3. The authors did the long-term stability of the domains. Since there may be a large number of proteases in the physiological environment of Clostridium difficile, it is very meaningful for the follow-up study of the candidate domains to explore the stability of the candidate domains for major proteases, or how to improve their ability to resist proteases.

Minor suggestions:

1. It is recommended to add a conclusion sentence at the end of the Discussion section, or to comment on the future application of the domain and the method for screening the domain.

2. This paper mainly focuses on the screening and biochemical characterization of candidate domains. The authors can change the keyword "C. difficile infection" to "C. difficile".

Round 2

Reviewer 1 Report

All my concerns have been answered satisfactorily